# Effect of Spiny-Cheek Crayfish (*Faxonius limosus*) on H$_2$O$_2$-Induced Oxidative Stress in Normal Fibroblast Cells

**Klara Zglińska** [1,*] **, Sławomir Jaworski** [2] **, Anna Rygało-Galewska** [1] **, Andrzej Łozicki** [1] **, Mateusz Roguski** [1] **, Magdalena Matusiewicz** [2] **and Tomasz Niemiec** [1]

1 Institute of Animal Sciences, Warsaw University of Life Sciences, Ciszewskiego 8 Street, 02-786 Warsaw, Poland
2 Institute of Biology, Warsaw University of Life Sciences, Ciszewskiego 8 Street, 02-786 Warsaw, Poland
* Correspondence: klara_zglinska@sggw.edu.pl

**Abstract:** *Faxonius limosus* (spiny-cheek) crayfish is an invasive species that is widespread in Europe. The aim of the study was to evaluate the possibility of using extracts from this crustacean as a source of natural antioxidants. For this purpose, selected bioactive compounds (total phenols, glutathione, vitamins A, E, D, β-carotene and amino acids) were characterised and the antioxidant activity of the extract was assessed in vitro and in relation to HFFF-2 fibroblast cells, in which oxidative stress was caused by the additive hydrogen peroxide (H$_2$O$_2$). The extract abolished the cytotoxicity of H$_2$O$_2$, decreased reactive oxygen species (ROS) secretion, increased cell viability and decreased the expression of caspase-3. The results suggest that *F. limosus* extract is a promising raw material with antioxidant properties.

**Keywords:** crayfish extract; spiny-cheek crayfish; oxidative stress; *Faxonius limosus*; antioxidant activity

## 1. Introduction

Oxidative stress arises as a result of an imbalance between the production of free oxygen radicals (ROS) and the organism's antioxidant capacity. ROS can also be formed as a result of the action of external factors (e.g., UV radiation) and during the defensive reactions of the body's immune system. The production of ROS also occurs during normal physiological processes in various cell compartments. Under physiological conditions, these processes are under the strict control of the organism, as a result of the action of enzymatic and non-enzymatic defence mechanisms. Unfortunately, in many situations, the natural mechanisms of ROS regulation are not able to stop the increasing oxidative stress. In such a situation, a great deal of multicellular damage occurs, such as DNA damage, enzyme inactivation, peroxidation of cell membrane lipids and structural changes in proteins.

Fibroblasts are cells that are widely distributed in the animal body. Fibroblasts have a structural function but also contribute to a variety of immune responses. Persistent oxidative stress and the associated inflammation in humans develop into a variety of fibrotic changes, which proves that the sustained activation of the immune system can lead to the severe dysfunction of fibroblasts. Their presence in the skin also makes them particularly vulnerable to environmental causes of ROS overproduction, such as solar radiation or mechanical damage. Consequently, it is important to protect them from oxidative stress. One of the most effective methods of eliminating excess ROS production is the use of antioxidants. Due to the high demand, new sources of antioxidants are constantly being sought.

Aquaculture is considered the fastest-growing food production sector in the world. Global seafood consumption (including shellfish and fish) is increasing faster than global population growth. Therefore, it can be used to ensure sustainable food production in the future. Aquatic animals are among the world's most important protein sources [1]. This is

primarily because, in addition to being an easily digestible protein with the desired amino acid composition, they contain micronutrients and essential unsaturated fatty acids, which cannot be found in the tissues of land animals. Moreover, crustaceans contain natural antioxidants (mostly carotenoids) and bioactive peptides [2]. Freshwater crayfish are a much lesser known source of food and natural antioxidants.

*Faxonius limosus* (also known as *Orconectes limosus* and spiny-cheek crayfish) is an invasive species that is widespread in Europe [3,4] and has superior reproductive performance, tolerance to environmental conditions and resistance to diseases compared to native crayfish. For this reason, its population continues to grow, displacing native species [5].

Śmietana et al. (2020) examined crayfish *Faxonius limosus* caught in Poland, as a new food raw material. Crayfish abdomen meat is edible, contains little fat and has high content of unsaturated fatty acids. It is also a good source of protein with a favourable amino acid index (CS) and provides valuable minerals (Ca, K, Mg, Na, P) [5]. Crayfish meat also has an interesting amino acid composition, distinguished by its high histidine content [6], which has antioxidant properties, such as free radical scavenging and iron (II) chelation. It is known that spiny-cheek crayfish contain numerous carotenoids in the exoskeleton and, to a lesser extent, in the muscles [7]. However, due to the low number of catches and the difficulty in separating the armour from the muscles, it is not yet used as a functional food source. However, it appears to be be a good source of natural antioxidants. Broader characterisation of the raw material and the development of a convenient processing method are needed.

This study aimed to investigate the antioxidant properties of *F. limosus* crayfish extract (FLE) in HFFF-2 fibroblast cells induced with hydrogen peroxide. To simplify the procedure and make it attractive to the industry, it was decided to use the whole crayfish, without separating the carapace from the meat.

## 2. Materials and Methods

### 2.1. Samples

Spiny-cheek crayfish individuals ($n = 40$) were caught from lakes near Warsaw. Immediately after this, the crayfish were placed in a tank filled with water. After being transported to the laboratory, the crayfish were frozen at $-80$ °C.

As crayfish belong to invertebrates, following EU and Polish law, the consent of the ethics committee was not required. In the process of obtaining research samples, efforts were made to avoid unnecessary stress and pain to the crayfish. The "Guidelines for the Treatment of Animals in Behavioural Research and Teaching" [8] were followed.

### 2.2. Extract Preparation

Extraction was performed by ultrasound-assisted maceration. Whole frozen crayfish were freeze-dried for 2 days and then ground until a homogeneous mass was obtained. The sample was then flooded with water (1:10), shaken for 30 min and sonicated in an ultrasound bath for 15 min. The extract was then centrifuged and filtered through a Whatman filter and lyophilised.

### 2.3. Extract Characterisation

2.3.1. Bioactive Compounds

The total phenolic (TC) content was determined using the Folin–Ciocalteu method, as before [9]. Diluted extract (0.5 mL) was mixed with Folin–Ciocalteu reagent (2.5 mL). After 3 min incubation, 7.5% $Na_2CO_3$ (2 mL) was added and samples were incubated in a water bath (50 °C, 15 min). The results were read spectrophotometrically at 750 nm. TC content was calculated using the calibration curve of quercetin and presented as quercetin equivalent milligram per gram (mg/g) of extracts $N = 6$.

The modified Ellman method was used to determine the glutathione content spectrophotometrically [10]. The procedure detects non-protein -SH groups in the reduction reaction of 5,5′-dithiobis (2-nitrobenzoic acid) (DTNB) by thiol compounds and the pro-

duction of coloured 2-nitro-5-mercaptobenzoic acid. Extract ($N = 6$) was homogenised with 0.1 M phosphate buffer pH 7.4 and centrifuged (3500 rpm, 12 min). The supernatants (1.5 mL) were mixed with 50% TCA (78.96 µL) and samples were centrifuged (2500 rpm, 5 min). Next, the sample (25 µL) was pooled with 0.2 M phosphate buffer pH 8.0 (200 µL) and with DTNB (25 µL). The absorbance was read on a microplate reader (Infinite M200; Tecan, Männedorf, Switzerland). The standard curve was obtained using different dilutions of GSH in 2.5% TCA. Analysis of β-carotene and vitamins A, D (group), K and E (α-tocopherol) was conducted using an Agilent 1100 high-performance liquid chromatography (HPLC) system (Agilent, Waldbronn, Germany). Separations were performed at 25 °C on the Zorbax Eclipse XDB C8 column (Agilent). Parameters: solvent flow rate—1.2 mL/min in a linear gradient of 90:10 (vol./vol.) methanol to water, injection volume—10 µL, temperature of the column oven—30 °C. Detection was performed with a UV–vis detector at 265 nm for β-carotene and vitamin D and at 230 nm for α-tocopherol. The standard solution was produced by dissolving each standard substance in methanol. Identification was carried out through comparison with standard substances (Sigma-Aldrich, Darmstadt, Germany) $N = 2$.

The crude protein was determined by the Kjeldahl method according to AOAC International, as before [11]. The amino acid content was determined by ultra-performance liquid chromatography. The Acquity UPLC System with a PDA detector was used, and detection was performed at wavelength 260 nm (Waters Corp., Milford, MA, USA) and using an AccQ-Tag Ultra C18, 1.7 µm, 2.1 × 100 mm column (Waters Corp., Milford, MA, USA). On the basis of the protein content of the extract, the results were converted into the content of the respective amino acid in the extract $N = 2$.

$$\text{Chemical score} = (\text{AA content in extract}/\text{AA content in FAO Proposed Pattern (1981)}) * 100$$

### 2.3.2. Antioxidative Activity In Vitro

The antioxidant activity of the crayfish extract was assessed using the DPPH oxygen radical reduction test and compared with the standard antioxidants gallic acid and beta-carotene. For this, 290 µL of 0.1 M DPPH was mixed with 10 µL of the extract or standard solution at a 100 µg/mL concentration. After 20 min in the dark, the results were read at a wavelength of 570 nm in a plate spectrophotometer (Tecan, Männedorf, Switzerland. The results are presented as % inhibition of the DPPH radical.

### *2.4. Cell Study*
### 2.4.1. Cell Line

Normal fibroblast cell lines (HFFF-2) were purchased from the European Collection of Authenticated Cell Cultures (ECACC) and maintained in Dulbecco's modified medium (DMEM, VWR Chemicals, Radnor, PA, USA) supplemented with 10% foetal bovine serum (FBS, EU origin, Biowest, Bradenton, FL, USA), 1% penicillin and streptomycin (VWR Chemicals), at 37 °C in an atmosphere of 5% $CO_2$/95% air in a NuAire DH AutoFlow $CO_2$ Air-Jacketed Incubator (Plymouth, MN, USA). The passages from 4 to 10 were used.

The research was conducted in the concentration range of 50–200 µg/mL FLE in DMEM. This was selected on the basis of preliminary studies conducted on fibroblast-like cells, of which the results were the basis for the registration of a patent for the preparation of *F. limosus* extract, registered in the Patent Office of the Republic of Poland under the number PAT.237977.

### 2.4.2. Light Microscopy

Cells were placed in 6-well plates. The next day, the medium (DMEM) was replaced with a new one with the addition of 100 µg/mL extract or the same amount of phosphate-buffered saline (PBS), 6 wells each. After 24-h incubation with hydrogen peroxide, half the wells were added with the extract and half with PBS. Thus, 4 groups were obtained:

1.   PBS only control, which received neither extract nor hydrogen peroxide;

2.　　　H$_2$O$_2$ only control, which received no extract but had added H$_2$O$_2$;
3.　　　100 μg/mL extract, which received the extract without H$_2$O$_2$;
4.　　　100 μg/mL extract, which received the extract and H$_2$O$_2$.

An inverted light microscope (Leica, TL-LED, Wetzlar, Germany) was used to assess HFFF-2 cell morphology over the next 12 h. The microscope was connected to a digital camera (Leica MC190 HD with LAS V4.10 software). Six pictures were taken of each sample.

### 2.4.3. Cytotoxicity Test

To measure the cytotoxicity, the LDH Assay Kit (Cayman) was used, according to the manufacturer's protocol. Cells were plated into 96-well plates at $10^4$ cells/well, and 200 μL DMEM was added to each well. The next day, the medium was replaced with DMEM low-serum (1%) with different extract concentrations: 50, 100 and 200 μg/mL or controls (PBS, LDH-positive and Triton-X). After 24 h, oxidative stress was induced by adding 500 mM hydrogen peroxide. Simultaneously, a control group receiving only PBS was used without adding hydrogen peroxide. After 12 h, the test was performed according to the manufacturer's procedure. The plate was centrifuged at $400 \times g$, and then the supernatant was transferred to a new plate and 100 μL of LDH reaction solution was added. After 30-min incubation at 37 °C, the absorbance was read at 490 nm (Tecan, m nano, Mennedorf, Switzerland).

### 2.4.4. ROS Secretion Test

The detection of reactive oxygen species was performed by 2′,7′-Dichlorodihydrofluorescein Diacetate Staining (DCFH-DA), according to the procedure developed by Kim and Xue [12]. The cells were handled in the same way as for the LDH test. At 12 h after the induction of oxidative stress, the DMEM was replaced with a fresh medium and 500 μL of the 10 μM DCFH-DA was added to each well. The assay plate was incubated for 30 min at 37 °C. Then, each well was washed once with neat DMEM and twice with PBS. The wells were then filled with 100 μL of PBS each and fluorescence intensity was measured using a fluorescence microplate reader (Tecan 200, Switzerland) at an excitation wavelength of 485 nm and an emission wavelength of 530 nm.

### 2.4.5. Proliferation Test

The XTT Kit (Sigma-Aldrich/Merck, Darmstadt, Germany), a colorimetric assay, was used to measure cell viability and proliferation. The assay is based on the extracellular reduction of XTT by NADH produced in the mitochondria. Cells were plated into 96-well plates at $10^4$ cells/well at a volume of 100 μL DMEM. The next day, 90 μL of fresh culture medium and 10 μL of extract at concentrations of 50, 100 and 200 μL/mL were added to each well. Oxidative stress was triggered in the same way as with the LDH test. After 12 h, the XTT solution (50 μL) was added to each well. The microplate was allowed to react for 1 h at 37 °C. Results were examined using a microplate reader (Tecan m nano, Switzerland), according to the manufacturer's protocol $N = 6$.

### 2.4.6. Caspase-3 Test

The fluorometric caspase-3 kit (BioVision, Milpitas, CA, USA) was used according to the manufacturer's protocol. The test allows the detection and quantification of caspase-3 activity by using a synthetic substrate, DEVD-AFC, which, upon cleavage by caspase-3, will emit a strong, stable fluorometric signal. $N = 4$. Cells were plated in 96-well plates and/or treated with the extract and then with hydrogen peroxide (except the PBS only control) as in previous tests. To assess caspase-3 activity, 100 μL of an assay buffer with the addition of a substrate was added to each well, followed by incubation at 37 °C for 1 h. Fluorescence was measured using a Tecan Infinite 200 microplate reader (Tecan, Durham, NC, USA) at Ex/Em = 400/505 nm.

*2.5. Statistics*

The data were analysed using one-way analysis of variance (ANOVA) with Tukey's post-hoc test and Statistica 13.3 software (TIBCO Software Inc., Palo Alto, CA, USA). Differences with a *p*-value $\leq 0.05$ were defined as statistically significant.

## 3. Results

*3.1. F. limosus Crayfish Extract Is a Source of Valuable Bioactive Substances*

*F. limosus* crayfish extract (FLE) contained phenols, glutathione and vitamins (A, E, D, K) in the amounts shown in Table 1. The main labelled components of the extract were phenols and glutathione. Fat-soluble vitamins and β-carotene were present in small amounts. Among the detected vitamins, the highest level was noted for vitamin E.

**Table 1.** Content of selected bioactive compounds in the extract of whole spiny-cheek crayfish.

| Compound | Content |
|---|---|
| Total phenols (µg/mL [1]) | 13.321 |
| GSH (nmol/mg) | 10.755 |
| β-carotene (mg/g) | 0.029 |
| Vitamin A (mg/g) | 0.085 |
| Vitamin E (mg/g) | 0.785 |
| Vitamin D (µg/g) | 0.084 |
| Vitamin K (µg/g) | 0.104 |

[1] expressed as quercetin equivalent.

The content of amino acids in the extract and their comparison with the reference protein (FAO, 1981) are presented in Table 2. High protein content characterised the extract, and, among the amino acids, histidine deserves special attention. The value in the extracted protein was higher than in the protein standard proposed by FAO. Thereafter, high chemical scores were recorded for lysine and threonine. Compared to the FAO reference protein, FLE was found to be the most deficient in isoleucine. In terms of weight, glutamine was the most abundant in the extract and the least was tryptophan.

**Table 2.** Amino acid (AA) content of *F. limosus* extract expressed in mg/g extract and chemical score (CS) of each AA.

| Compound | mg/g Crayfish Extract | CS |
|---|---|---|
| His | 28.91 | 128.65 |
| Ile | 27.94 | 50.33 |
| Leu | 60.75 | 65.66 |
| Lys | 61.78 | 91.65 |
| TSAA | 28.77 | 83.73 |
| TAAA | 75.24 | 77.99 |
| Thr | 38.75 | 83.77 |
| Trp | 9.83 | 67.64 |
| Val | 41.77 | 65.83 |
| Pro | 29.77 | - |
| Glu | 101.72 | - |
| Crude protein | 756.60 | X |

*3.2. F. limosus Crayfish Extract Has Antioxidant Properties Comparable to β-Carotene*

The extract showed high antioxidant activity, which did not differ significantly from the activity of β-carotene (Figure 1). The highest value was obtained for gallic acid, which is known to have high antiradical activity and is often used as a standard in antioxidant testing.

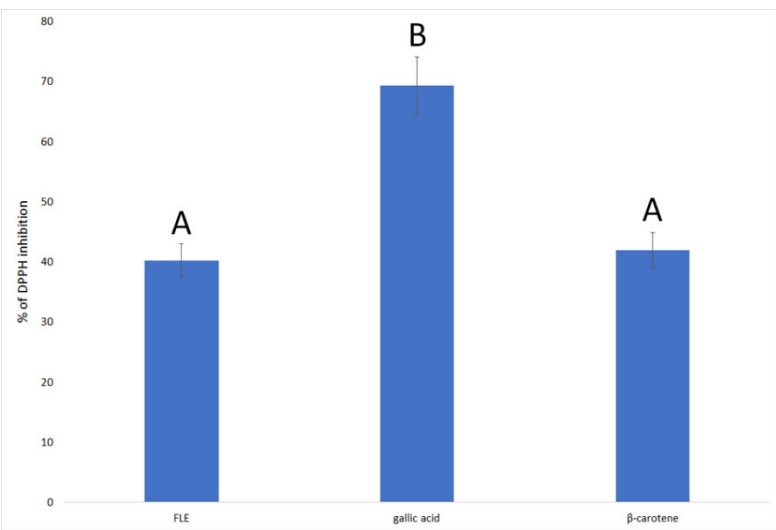

**Figure 1.** Antioxidant activity of *F. limosus* crayfish extract compared to gallic acid and β-carotene, expressed as % of DPPH inhibition. Different letters mean that the values for the bars differ statistically ($p < 0.05$), $N = 6$.

### 3.3. Effect of Extract on $H_2O_2$-Induced Oxidative Stress in Normal Fibroblast Cells

The addition of hydrogen peroxide clearly changed the typical morphology of HFFF-2 fibroblasts (Figure 2A). The cells became more oblong in comparison to the controls. Cell numbers were considerably reduced and more dead (detached) cells appeared. Earlier incubation with FLE (100 μg/mL) partially reduced this effect, but most cells did not have the typical HFFF-2 cell morphology (2b). The number of cells was higher and fewer dead cells were observed (Figure 2B). Application of the extract in cells without induction with hydrogen peroxide did not change the typical cell morphology (Figure 2D).

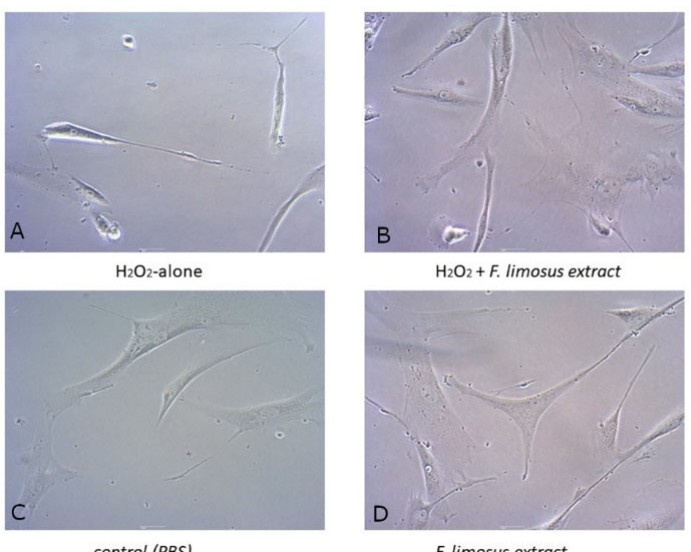

**Figure 2.** Morphology of cells treated with *F. limosus* extract or PBS and/or $H_2O_2$. (**A**) Cells treated only with 100 uM/mL hydrogen peroxide. (**B**) Cells treated with 100 ug/mL crayfish extract and 100 uM/mL hydrogen peroxide. (**C**) Control cells treated only with PBS. (**D**) Cells treated only with 100 ug/mL crayfish extract. Scale bar: 1 μm.

The results of the LDH test showed that the addition of a crayfish extract at the concentration of 50–200 μg/mL significantly reduced the cytotoxicity of hydrogen peroxide in HFFF-2 fibroblast cells (Table 3). The FLE concentration of 100 μg/mL was found to be the most effective, which caused the cytotoxicity in cells induced with hydrogen peroxide

to be lower than in cells receiving PBS (not induced with $H_2O_2$). In the case of the highest dose of the extract (200 µg/mL), the cytotoxicity was low but significantly higher than for the 50 and 100 µg/mL dose. This indicates that the concentration of 200 µg/mL is too high for HFFF-2 cells. Then, we planned to investigate selected mechanisms that may be responsible for this.

**Table 3.** Percentage of cytotoxicity in HFFF-3 cells induced with $H_2O_2$ after crayfish (*F. limosus*) extract treatment. Control—non-induced cells, treatment only with PBS; $H_2O_2$ only—cells induced with $H_2O_2$, treatment only with PBS; Crayfish extract 50–200 ug/mL—induced with $H_2O_2$ and treatment with different concentrations of crayfish extract.

| Group | % of Cytotoxicity | $p < 0.05$ |
|---|---|---|
| Control (PBS only) | 0.187 | A |
| $H_2O_2$ only | 43.925 | B |
| FLE (50 µg/mL) + $H_2O_2$ | 0.639 | A |
| FLE (100 µg/mL) + $H_2O_2$ | 0.019 | A |
| FLE (200 µg/mL) + $H_2O_2$ | 2.045 | C |

First, the release of ROS was investigated in the same model (Figure 3a,b).

The highest expression of ROS was observed in cells induced with hydrogen peroxide without the addition of an extract. Adding the extract in the amount of 100 µg/mL abolished the effect of hydrogen peroxide. Concentrations of 50 and 100 µg/mL, used without adding hydrogen peroxide, lowered ROS production below the values of cells treated with PBS only. At the extract concentration of 200 µg/mL, a trend towards higher ROS expression was observed than for the concentrations of 50 and 100 µg/mL, but this difference was not statistically significant. However, when combined with the LDH assay, where the 200 µg/mL dose was less effective in reducing the cytotoxicity of $H_2O_2$, this indicates that the dosage may be too high for HFFF-2 cells.

Then, it was decided to check whether the decrease in cell death after applying the extract was also due to the increase in their proliferation. The results of the proliferation test are shown in Figure 4.

Overall, the incubation of healthy cells with FLE increased their viability many times over. The strongest effect was demonstrated for the concentration of 100 µg/mL, which increased the number of viable cells by 6.3 times compared to the control incubated with PBS.

The addition of hydrogen peroxide alone reduced cell viability by 63%. In the oxidative stress model, the addition of *F. limosus* extract increased the viability of cells above the control value (PBS only). This effect was proportional to the concentration of the extract. The highest number of viable cells was observed after the incubation of cells with FLE at a concentration of 200 µg/mL. The cell numbers were 12.32 times greater than those for cells treated with hydrogen peroxide without incubation with FLE and 4.56 times greater than the control cells, which were not subjected to oxidative stress.

Then, the expression of caspase-3, a marker of early/moderate apoptosis, increased significantly after treatment of the control cells with hydrogen peroxide (Figure 5). Earlier application of *F. limosus* extract (100 ug/mL) reduced the caspase-3 expression in cells as compared to cells treated with hydrogen peroxide without the extract. Although the results were statistically significant, it should be noted that caspase-3 activity after stimulation with hydrogen peroxide only increased by 32.13%. Incubation with FLE prior to the induction of oxidative stress decreased caspase-3 activity by 21.42%. This indicates that this mechanism is not the main factor responsible for the protective effect of the extract against oxidative stress. The stimulation of HFFF-2 cell proliferation by FLE seems to have a much greater effect.

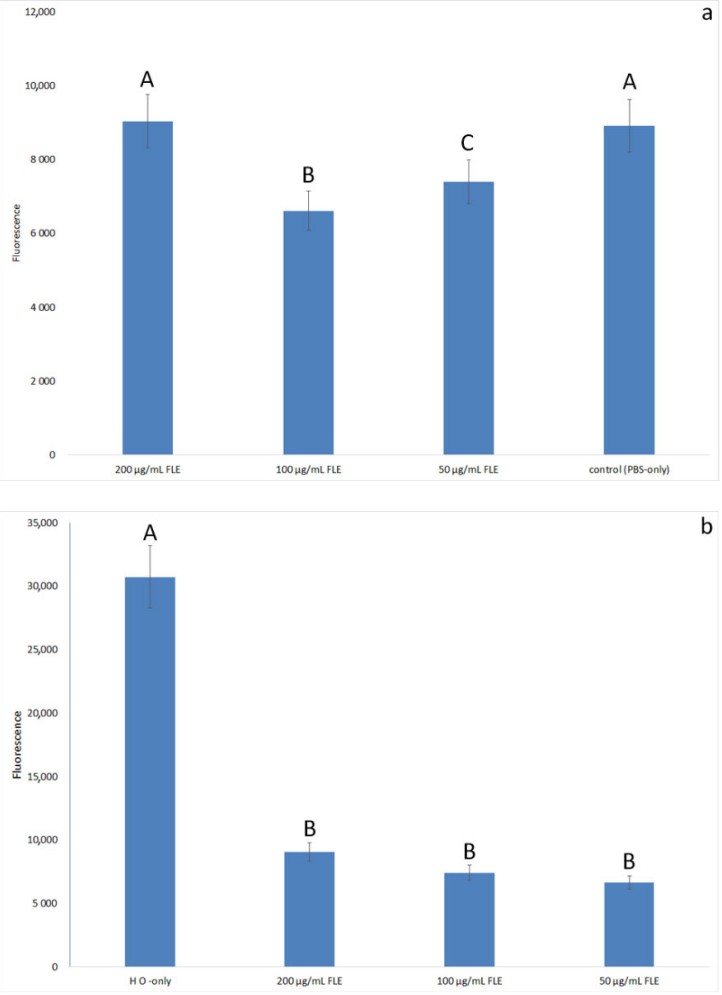

**Figure 3.** ROS expression (fluorescence) of HFFF-2 cells: (**a**) treated with 50–200 μg/mL *Faxonius limosus* extract (FLE) or PBS (control); (**b**) treated with 50–200 μg/mL *Faxonius limosus* extract (FLE) or PBS (control) and 500 μM hydrogen peroxide. Different letters mean that the values for the bars differ statistically (*p* < 0.05).

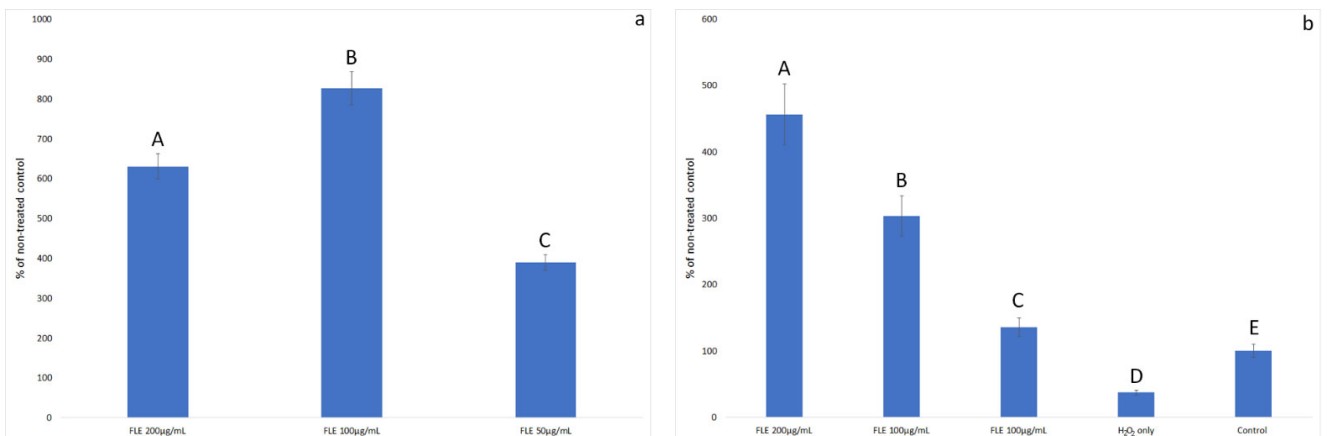

**Figure 4.** The proliferation of HFFF-2 cells treated with (**a**) Faxonius limosus extract and (**b**) treated with Faxonius limosus extract or PBS (only $H_2O_2$) and 500 μM hydrogen peroxide. Results are shown as percentage of control viability to which only PBS was added (no extract and no $H_2O_2$). Different letters mean that the values for the bars differ statistically (*p* < 0.05).

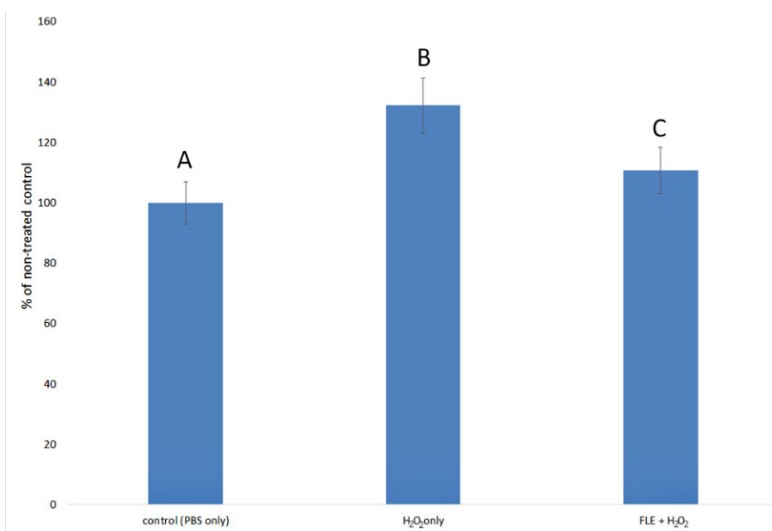

**Figure 5.** The expression of caspase-3, an early/moderate apoptosis marker, in HFFF-2 fibroblasts induced by hydrogen peroxide is shown as a percentage of control values (PBS only). $H_2O_2$ only—cells treated with $H_2O_2$ without the use of an extract; FLE + $H_2O_2$—cells incubated with 100 µg/mL *F. limosus* extract and 500 µM hydrogen peroxide. Different letters mean that the values for the bars differ statistically ($p < 0.05$).

## 4. Discussion

In the article, *F. limosus* crayfish extract (FLE) was characterised for the first time in terms of its many bioactive ingredients (vitamins, carotenoids, glutathione, minerals, amino acids), and it was estimated that it had antioxidant properties comparable to β-carotene. Due to the preparation method (water extraction), FLE contained few fat-soluble ingredients. The main antioxidant substances were phenolic compounds, which probably came from algae consumed by crayfish. Phenols are a broad group of compounds of great importance as antioxidants from natural sources. They come primarily from plant sources, but they can accumulate in the tissues of crustaceans. The antioxidant activity of phenols is based on the transfer of a single electron and hydrogen atom [13].

What distinguishes crustaceans from most plant-based antioxidant sources is their high protein content. The antioxidant effect of the extract was undoubtedly influenced by the relatively high content of GSH. The content of GSH in the extract was 10.76 nmol/mg and was comparable to that of shrimp *Palaemonetes pugio* embryo (10 embryo stage) [14] and ten times higher than in the water extract of Saposhnikovia divaricata root [10]. In addition, *F. limosus* crayfish extract protein has an amino acid composition similar to fish. It is characterised by, among others, relatively high histidine content. Compared to bivalve molluscs, *F. limosus* extract contained more histidine than *A. broughtonii* and *S. sachalinensis*, but less than *C. japonica* tissues. This amino acid is an essential amino acid (EAA) for mammals, poultry and fish. Due to the low supply in the diet, deficiencies are often encountered. Histidine deficiency is associated with impaired antioxidant protection of the body and affects many life processes, e.g., in rats, histidine supplementation provided neuroprotection at an early stage; in fish, it protected against glaucoma; and in humans, it helped to reduce the effects of metabolic syndrome. Moreover, histidine deficiency limited animal growth and milk synthesis in ruminants [15].

The antioxidant properties of FLE were at the level of β-carotene but were significantly lower than those of gallic acid. The lyophilisate from fermented shrimp waste inhibited 40% of the DPPH radical at a concentration of 1 mg/mL; in the reported study, the extract from *F. limosus* showed similar activity at a 10 times lower concentration [16]. However, many plant extracts have a stronger ability to inhibit the DPPH radical. For example, Proestos et al. (2012) studied extracts of popular Greek plants, of which *Eucalyptus globulus*, *Sideritis cretica*, *Origanum vulgare*, *Phlomis cretica*, *Phlomis lanata*, *Nepeta melissifolia* and *Mentha pulegium* had

higher antiradical activity than FLE [17]. Research on natural products must focus on the various mechanisms that may affect the protective effect against oxidative stress.

One of them is probably the action of glutathione, contained in the *F. limosus* extract. The biological functions of GSH in cells are related to the activity of the hydrosulphide group, thanks to which it participates in antioxidant and detoxification reactions. Numerous studies have demonstrated the function of endogenous glutathione in protective mechanisms against oxidative stress in the body. It has also been shown that GSH supplementation could reduce oxidative stress and inflammation markers in healthy volunteers [18] and HIV-infected patients [19]. It is also known that plant substances may have various effects on the antioxidant activity of glutathione, e.g., galangin and isoramnetin antagonise the action of GSH [20], while methyl-ether derivatives of cyclitols (D-pinitol and L-quebrachitol) supported this action [21].

Next, the protective properties of FLE against fibroblasts, in which the addition of hydrogen peroxide caused oxidative stress, were examined. The extract did not change the morphology of healthy fibroblast cells, and when used before induction with $H_2O_2$, the morphology was more similar to that of normal cells. *F. limosus* extract reduced the cytotoxicity of the hydrogen peroxide used. At a concentration of 100 µg/mL, the cytotoxicity of hydrogen peroxide was abolished entirely. Similar effects have been found for other aquaculture animals, e.g., protein isolated from mussels [22], oysters [23] and seahorses [24]. Moreover, the water extract from the tissue of the pen shell (Atrina pectinate) reduced cell death and ROS secretion caused by oxidative stress, but the effect was weaker than that of FLE [25]. Fatmy and Hamdi (2011) showed that the aqueous extract of *Procambarus clarkii* freshwater crayfish bodies decreased the markers of oxidative stress (MDA) and also increased the activity of antioxidant enzymes in the sera and livers of rats following CCl4 intoxication [26]. Similarly, in the reported study, it was shown that the addition of FLE reduced the release of ROS. Fibroblast cells incubated with FLE at 50 and 100 µg/mL before induction with hydrogen peroxide secreted ROS at the same level as the control cells (PBS only). The level of ROS secretion in cells that had not received an extract before the administration of hydrogen peroxide increased more than threefold.

It has also been shown that preincubation with FLE reduces the expression of caspase-3, a marker of early apoptosis, in $H_2O_2$-induced HFFF-2 cells. The activity in lowering the apoptosis of fibroblasts was demonstrated for the polyphenol-rich strawberry extract [27], as well as for silk lutein [28] and astaxanthin [29]. It should be noted, however, that the low increase in caspase-3 expression after the induction of oxidative stress indicates that this mechanism has little effect on HFFF-2 cell death following $H_2O_2$ treatment. The FLE effect itself, although statistically significant, only slightly decreased the expression of caspase-3.

The most promising mechanism of action of the FLE is to increase the proliferation. The presented study showed that the extract added to healthy fibroblast cells increased the number of viable cells in relation to cells treated only with PBS. The addition of hydrogen peroxide reduced the cell viability by over 60%. Adding the extract before the induction of oxidative stress completely eliminated the effects of death caused by hydrogen peroxide, and the protein fraction of the extract was likely responsible for this result. FLE has high lysine content, and this essential amino acid is one of the most important building blocks of many proteins in the body. It has long-known properties supporting tissue regeneration after injuries. In in vitro models, lysine enhanced the production of growth factors by fibroblasts. It also has the ability to induce strong inflammatory and immune responses (both humoral and cell-mediated), which could also have influenced the antioxidant effect of the FLE on HFFF-2 cells [30]. Moreover, proline is an amino acid proven to increase the regeneration of connective tissue after injuries. It is an essential component of collagen and also plays an important role in protein synthesis and structure and metabolism, as well as wound healing, antioxidant reactions and immune responses [31,32].

Phenols present in the extract may also have been responsible for the increase in proliferation. It was shown that phenolic-rich plant extracts significantly increased the proliferation of normal fibroblasts [33,34].

It is clearly visible that the concentration of FLE 200 μg/mL acted differently on normal cells and cells subjected to oxidative stress. This dose in healthy cells had a significantly lower pro-proliferative effect than 100 μg/mL. Contrary to lower concentrations, the concentration of 200 μg/mL FLE did not lower ROS expression below the physiological level, and it did not protect the cells significantly from the cytotoxicity induced by hydrogen peroxide. As is known, most antioxidants administered in too high concentrations show a pro-oxidative effect. Such an effect may be caused by, among others, some phenols [35], ascorbic acid [36] and β-carotene [37]. It has been shown that supplementation with high doses of β-carotene leads to a reduction in the amount of a-tocopherol and an increase in glutathione oxidation in cells from concentrations of 10 μM. High doses of β-carotene also impair the natural antioxidant systems by influencing antioxidant enzymes [37]. A similar effect was also observed for some phenols. Acting pro-oxidatively, they can limit cell proliferation and even lead to apoptosis. FLE is a complex mixture that also includes minerals. In the presence of oxygen, iron (Fe) and copper (Cu) catalyse the redox cycle of antioxidants, resulting in the formation of oxidants that can damage DNA, lipids and other biological molecules of cells [35]. In addition, the lack of an effect of the highest dose of FLE on the reduction of ROS expression in healthy fibroblasts, despite the strong effect of lower doses, may also result from a change in pH. The addition of an alkaline extract to the neutral environment of the cell medium caused a dose-dependent decrease in pH. It was shown that, in an alkaline environment, the antioxidant activity of phenols decreases and that extracts with higher content of phenols had a stronger pro-oxidative effect at an alkaline pH [38]. Conversely, after the induction of oxidative stress, the pro-proliferative effect was dose-dependent at concentrations of 50–200 μg/mL. This is probably due to the fact that the antioxidant requirements of cells increased under oxidative stress. In the future, research is planned to determine the optimal dose and understand the pro-proliferative effects of FLE.

## 5. Conclusions

In this study, for the first time, the water extract of the freshwater crayfish *F. limosus* was characterised and was shown to protect fibroblast cells against $H_2O_2$-induced hydrogen peroxide. FLE significantly decreased ROS expression, decreased the toxicity of hydrogen peroxide and increased proliferation in cells subjected to antioxidant stress. In light of these findings, FLE may represent a new source of natural antioxidants with applications in the food, cosmetic and medicine industries. In addition, the method of preparing the extract is simplified, and the high availability of the hitherto unused raw material favours optimism for its application potential. Further research is needed to understand other mechanisms responsible for the protective effect of FLE in relation to fibroblasts in a state of oxidative stress, as well as to optimise the dose of the extract.

## 6. Patents

The method of preparing the extract was patented by the Patent Office of the Republic of Poland under the number PAT.237977.

**Author Contributions:** Conceptualisation, K.Z., A.Ł. and T.N.; Data curation, K.Z.; Formal analysis, K.Z., A.R.-G., M.R. and M.M.; Investigation, S.J.; Methodology, K.Z., S.J. and M.M.; Project administration, K.Z.; Resources, S.J. and T.N.; Supervision, T.N.; Validation, K.Z.; Visualisation, A.R.-G. and M.R.; Writing—original draft, K.Z.; Writing—review and editing, A.Ł. and T.N. All authors have read and agreed to the published version of the manuscript.

**Funding:** This research received no external funding.

**Institutional Review Board Statement:** Not applicable.

**Informed Consent Statement:** Not applicable.



**Data Availability Statement:** The data presented in this study are available on request from the corresponding author.

**Conflicts of Interest:** The authors declare no conflict of interest.

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
