# Peer review of "Effect of Spiny-Cheek Crayfish (Faxonius limosus) on H2O2-Induced Oxidative Stress in Normal Fibroblast Cells"

_applsci, doi:10.3390/app12178546_

Round 1

Reviewer 1 Report

The manuscript ‘Effect of Spiny-Cheek Caryfish (Faxonius limosus) on H2O2-induced oxidative stress in normal fibroblast cells’ report the antioxidant effect of Faxonius limosus crayfish extract (FLE) using H2O2-induced HFFF-2 fibroblast cell model. However, there are critical concerns and problems on the manuscript. So, the reviewer is unable to agree with the acceptance of manuscript with the current form. The manuscript needs significant upgrade including the presentation of the figures.

Some comments are as follows.

The introduction should describe why this study is important and needed in a broad context. However, the authors simply mentioned that since Faxonius limosus crayfish is invasive species widespread in Europe and is not used as a functional food source, the aim of this study was to evaluate the extract as a good source of natural antioxidants. The authors should describe to be more logical and detail.

The materials and methods should be described with sufficient detail to allow others to replicate and build on published results. However, the detailed experiment conditions are not described in the materials and methods (ex., Extract preparation).

The authors should explain the logical reasons on the selected cell type and FLE concentration.

In Figure 3, 100 ug/mL FLE treatment to HFFF-2 cells has approximately 150% higher proliferation level than FLE 200 ug/mL treatment group. However, FLE 200 ug/mL treatment to oxidative stress(H2O2)-induced cells has more proliferation effect than 100 ug/mL treatment group. The authors need to explain these differences in the discussion.

In Figure 4, although the treatment of H2O2 to HFFF-2 cells increased the expression of casepse-3 up to approximately 130%, FLE treatment to oxidative stress-induced HFFF-2 cells decreased the expression of caspase-3 to only 110%. Rather, FLE remarkably increased the proliferation of oxidative stress-induced HFFF-2 cells up to 450%, compared to control (Figure 3). Therefore, it seems that FLE has the proliferative effects, not the inhibition of apoptotic effect. The authors need to focus on the proliferation effect, not a apoptotic effect.

Author Response

Dear Reviewer,

Thank you for your comments, which helped enrich the manuscript.
The introduction has been developed to explain the need for research, its purpose, and the type of cells selected. The method of choosing the extract concentration is described in section 2.4.1.
The methods have been completed.

We tried to describe the unusual effect of the 200ug / ml dose of FLE on cell proliferation in the discussion from line 356.

The discussion was changed in such a way as to emphasize the pro-proliferative role of the extract further. The low impact on apoptosis was also raised, as indicated by the reviewer (from line 331).

The article has been thoroughly reviewed and improved in terms of content and typography. The figures have been recreated, and the manuscript was also submitted for corrections to an English native speaker.

Thanks again for your valuable comments. I hope that in this form, the new version of the manuscript will be recommended for publication.

Reviewer 2 Report

Valuable work has been done and interesting results have been obtained. The objective and hypothesis of the article are not well stated in the introduction. It can be accepted after the corrections are made.

the most important points are listed below, which include:

-          Although the English is satisfactory in general, there are some sentences and words that have no sense. A revision of the English is mandatory.

-          MS needs to be reviewed by the author to completely eliminate writing and typographical errors.

-          The results section needs to describe the results well. The results should be reviewed.

-          Figures should be redrawn and horizontal lines removed and drawn in a more appropriate format.

-          Error bars (SD or SR) in charts should be added.

-          In the discussion, focus only on your own results.

-          Conclusion: is interesting, if the authors focused on the novelty of the data for improving the mechanisms by which tillage practice is able to improve crop yield and soil physical properties it will be better.

Author Response

Dear Reviewer,

Thank you for your comments, which helped enrich the manuscript.
The introduction has been developed to explain the need for research, its purpose, and the type of cells selected.

  1. The manuscript was also submitted for corrections to an English native speaker.
  2. The article has been thoroughly reviewed and improved in terms of content and typography.
  3. The description of the results has been expanded.
  4. The figures have been recreated.
  5. The discussion was expanded. A more detailed description of own results has been introduced. Some passages that focused too much on the results of other authors have been removed.

Thanks again for your valuable comments. I hope that the new version of manuscript will be recommended for publication in this form.

Reviewer 3 Report

It's a good idea to obtain natural antioxidants from Aquatic animal like Faxonius limosus (Spiny-Cheek) crayfish, which is invasive species widespread. This research showed that  F. limosus crayfish extract contains valuable bioactive substances and these substances have antioxidant properties. The authors verified these properties using H2O2-induced oxidative stress in normal fibroblast cell. Overall, the design and data is interesting. A few minor issues need to be improved before accept for publication.

1. Check the formulas like H2O2, CO2 etc. throughout the whole manuscript.

2. Give the full name and then abbreviations of the terms like ROS throughout the manuscript including Abstract. Use all the abbreviations correctly.

3. Check "3.3 Effect on extract on ".

4. The data will be more stronger if the authors provide protein assay of the cell proliferation and apoptosis experiments.

Author Response

Dear Reviewer,

Thank you for your comments, which helped enrich the manuscript. The article has been thoroughly reviewed and improved in terms of content and typography. 
1 & 2. The article has been corrected regarding the correct use of formulas and abbreviations.
3. Section 3.3. has been checked and expanded. In the results, captions a, b, c, and d were added to the figure.
4. Thank you very much for the suggestion of performing apoptosis and proliferation tests. We plan to make a Protein assay in the next project. We want to focus here on the mechanisms that make cells more viable after treatment with the extract. We have planned a larger experiment during which we will perform both the apoptosis / necrosis test, the Brdu proliferation test, as well as protein growth factor arrays. This is another project that will not address hydrogen peroxide but the pro-proliferative properties themselves, so we do not include this research in this article.

I hope that my explanations will prove to be sufficient and that the new version of the manuscript will be approved for publication.

Round 2

Reviewer 1 Report

The manuscript is improved but needs more efforts to be published. The authors did not respond point by point for the reviewer questions. The graphic of Figures quality is poor and inappropriate for publication. Figure 1 is missing in the revised manuscript. To say there is antiapoptitic properties, additional experiments are needed. Caspase-3 expression alone is not enough. For figure 2 cell pictures, additional pictures that shows many cells are needed. The current cell pictures are too much closed up. Also, a scale bar is needed in the picture.

Author Response

Thank you for the following comments that helped us enrich the article. 

1) The graphics were created in higher resolution and had a quality of 300dpi. In the latest version of the manuscript, we have included graphs separately from the manuscript to reduce the possibility of a loss of quality.

2) Thank you for the note regarding the absence of Figure 1. A revised version of this graphic has been added with the others.

3) A paragraph discussing caspase-3 expression was added at the request of another reviewer. Of course, these results are indeed too weak to be related to an anti-apoptotic effect. The manuscript has been revised not to suggest an action towards apoptosis. Please check lines 254-261 and 325-330 in the new manuscript. Due to the uninteresting results of caspase-3, we decided not to investigate further for apoptosis. Currently, a new study focuses on the pro-proliferative properties of F. limosus extract.

4) In Figure 2 the 40x approximation was used because our main goal was to show that F. limosus extract did not adversely affect cell morphology. We tested the number and viability of HFFF-2 cells primarily with the xtt test, which assessed the number of metabolically living cells. The microscope that I used to take the images shown in Figure 2 does not have a 10x objective. We have such a lens in an old microscope, which has a much worse quality of the photos taken, and I attach photos taken at a magnification of 10x from a worse microscope. Due to their quality, I suggest not adding them to the central part of the manuscript but putting them in a supplementary file.

5) The scale bar is marked in the photos. It is automatically added by the microscope software. In fact, however, it was hardly visible. The bar has been bolded to improve readability.

As for the English language, it was sent to a professional proofreading company before sending the revised manuscript after the first review round. The applied linguistic changes were visible in blue in the previous version. Proofreading was carried out by a person from Great Britain.

I hope that my explanations and introduced changes will prove to be sufficient and that the manuscript will be accepted for publication. Of course, I stay in touch when needed.

Reviewer 2 Report

Corrections were duly made by the authors. In my opinion, the article can be accepted in the current format.

Author Response

Thanks again for your valuable comments. I am glad that you accepted the article in this form.